# CO Detection System Based on TDLAS Using a 4.625 μm Interband Cascaded Laser

**DOI:** 10.3390/ijerph191912828

**Published:** 2022-10-07

**Authors:** Kun Li, Boyang Wang, Mingyao Yuan, Zhixiong Yang, Chunchao Yu, Weijian Zheng

**Affiliations:** 1Kunming Institute of Physics, Kunming 650223, China; 2Yunnan Normal University, Kunming 650029, China

**Keywords:** gas detection, carbon monoxide, remote measurement

## Abstract

During industrial operations and in confined places, carbon monoxide (CO) may collect in harmful proportions if ventilation is insufficient or appliances are not properly maintained. When the concentration of CO is too high, it might result in suffocation, coma, or even death. The detection of tiny concentrations of CO plays an important role in safe production. Due to the selective absorption of specific wavelengths of light by gas molecules, lasers have a wide range of applications in the field of gas detection. In this paper, a tunable diode laser absorption spectroscopy (TDLAS) system for CO detection was constructed using an interband cascaded laser (ICL) with a central wavelength of 4.625 μm. The modulated signal generated by the FPGA module was output to the laser controller to modulate the laser. The signal received by the detector was input to the FPGA module. After lock-in amplification, the second harmonic signal of high frequency modulation was output. Several concentrations of CO that were dispersed via static gas distribution were identified. A CO detection system with an open optical path was constructed, and the detection distance was about 8 m. The minimum detectable concentration is around 10.32 ppmm. The concentration of CO in the open optical path was 510.6 ppmm, according to the calibration of the detected concentration. The remote detection system based on TDLAS using an ICL can be used to monitor CO in the open optical path.

## 1. Introduction

Incomplete combustion of carbon fuel produces carbon monoxide (CO) during industrial processes such as coking and steelmaking, as well as in enclosed spaces such as mines and submarines. Suffocation, unconsciousness, or even death can occur when CO concentrations are exceeded [1]. The temperature of the coal seam has a direct impact on the amount of CO that is produced during coal mining. It commonly serves as an indicator gas for a coal seam fire warning [2]. The main methods of detecting CO are electrochemical test, catalytic combustion, and laser detection [3]. The electrochemical approach relies on the electrical signal generated by the chemical reaction of the target gas and the sensor electrode; in addition, it is easily affected by other gases. Through the process of catalytic combustion, heat is produced. With the increasing temperature, the detection electrode wire’s resistance increases. It is typically used to detect flammable gases. Laser detection is based on the selective absorption of light by gas molecules, which has the advantages of fast detection and low detectable concentration.

In order to determine the concentration of the detected gas, laser direct absorption detection technique measures the attenuation of a certain laser over a set length of gas [4,5,6,7,8]. The tunable diode laser absorption spectroscopy (TDLAS) system uses a wavelength-tunable semiconductor laser to obtain gas absorption spectra within the tuning range. The modulated laser is incident on the detector after passing through the gas, and the concentration of the target gas is proportional to the harmonic signal intensity of the high-frequency modulation signal. Usually, the second harmonic signal extracted by the lock-in amplifier is used as the detection signal. The concentration of the gas to be measured is obtained through the calibration of the standard concentration gas [9,10,11,12,13]. Some of the disadvantages associated with the laser direct absorption detection technique are a low signal-to-noise ratio, large amount of data calculation, etc. Many of them can be improved with TDLAS use. Semiconductor diode lasers found immediate application in the 1970s as much needed tunable sources for high-resolution laser spectroscopy commonly referred to as TDLAS. In reference [14], available semiconductor lasers for spectroscopy in the near- and mid-infrared spectral region have been reviewed together with the main features of TDLAS. A theoretical description of the wavelength-modulation spectrometry technique is given by Kluczynski P et al. [15]. Chen et al. [16] introduced an instrument that takes advantage of a mid-infrared quantum cascaded laser (QCL) operating at 4.8 μm and a mercury cadmium telluride (HgCdTe) mid-infrared detector. Low detection sensitivity down to 50 nmol/mol level in 4 s acquisition time was achieved using a multipass cell with a 76 m absorption path length. The simultaneous atmospheric pressure measurement of the trace gases methane (CH_4_) and CO using an open-path sensor based on TDLAS has been described. The detection limit of 0.58 parts per million by volume (ppmv) for CO and 0.4 ppmv for CH_4_ was accomplished at 1 s averaging time by using a distribution feedback (DFB) laser operating at 2.33 μm [17]. Shao et al. [18] adopted a DFB laser to constantly monitor CO and CH_4_ in an atmosphere based on TDLAS. The absorption signals of the sample gases were improved using a multipass absorption cell with a 72 m optical path length.

The laser detection of CO is typically performed in a multipass absorption cell. Gas detection in an open environment is typically necessary in practical applications. The telemetry detection of CO in an open optical path has received relatively little attention. CO has a substantially higher absorption coefficient at 4.6 μm than it does at 2.3 μm. The interband cascaded laser is currently the dominating light source for gas detection in the infrared spectrum. It benefits from room temperature functioning, a small linewidth, and an adjustable wavelength. In this paper, a TDLAS telemetry system is constructed using an interband cascade laser with a center wavelength of 4.625 μm. To accurately identify low concentrations of CO, the gas concentration is measured in accordance with the concentration calibration. The detection device can be employed in industrial parks, coal mining sites, and restricted places for fixed-point monitoring of low-concentration CO.

## 2. TDLAS Detection Principle

The light intensity is attenuated by the gas being measured in TDLAS, which uses a laser to emit a specific wavelength of light that is absorbed. The gas concentration directly relates to the degree of light intensity attenuation. It adheres to the Lambert-Beer law [19].
(1)$$I[v(t)]=I_{0}[v(t)]\exp\{-\alpha [v(t)]\mathrm{cL}\},$$
where I_0_ is the intensity of incident laser, and I is the intensity of transmitted laser α is the absorption coefficient of the gas, and V is the frequency of the laser. c is the gas concentration, and L is the light path length. The direct absorption technology detected can reversely calculate the concentration of the gas by the light intensity after attenuation.

When detecting trace gases, wavelength modulation technology can significantly reduce noise, boost the signal-to-noise ratio, and lower the detection threshold. The instantaneous frequency of laser emission at this time is
(2)$$v(t)=v_{0}+\sigma_{v}\cos(2\pi \mathrm{ft}),$$
where $v_{0}$ is the selected laser center frequency, $\sigma_{v}$ is the modulation signal amplitude, and f is the modulation frequency.

The Fourier series expansion of Equation (2) can be obtained
(3)$$I\left(v_{0},t\right)=\overset{\infty }{\underset{n=0}{\sum }}A_{n}\left(v_{0}\right)\cos(n2\pi ft),$$

The attenuation signal after gas absorption is received by detector and the second harmonic signal is obtained after phase-locked amplification [20]
(4)$$A_{2}\left(v_{0}\right)=\frac{I_{0}cL}{4}{\sigma_{v}}^{2}\frac{d^{2}\sigma \left(v\right)}{dv^{2}}{|}_{v=v_{0}},$$

Therefore, the concentration of the gas to be measured can be inverted by the direct current component I_0_ and the second harmonic amplitude
(5)$$c=\frac{A_{2}\left(v_{0}\right)}{I_{0}\mathrm{KL}},$$
where K is the calibration constant. Equation (5) shows that the gas concentration and second harmonic amplitude have a linear relationship. The determined gas concentration can be derived by inversion through standard gas calibration.

The selection of spectral lines is critical for gas measurement using TDLAS technology. The absorption spectra of CO and NO were shown in Figure 1. According to the HITRAN2012 spectral library, the pressure was 1 atm and the temperature was 296 K. In order to reduce the interference of N_2_O on CO detection, a laser source with a central wavelength of 4.625 μm was selected.

## 3. Experimental Equipment

The CO telemetry system based on TDLAS was shown in Figure 2. The light source was a quantum cascade laser with a central wavelength of 4.625 μm. At a constant temperature, the laser source was tuned around 15 nm by current, with a linewidth of less than 2.14 × 10^−4^ nm. When the working temperature was 20 °C and the input current of the laser was 72 mA, the output laser wavelength was 4.625 μm and the output power was 7.9 mW. The diameter of the laser was 3 mm, and the divergence angle was 35° × 55°. After being collimated by the collimation mechanism, the spot size at 3 m was 4.5 mm. About 30% energy was lost after the collimator. The light reflected from the corner reflector traveled via the focusing lens before concentrating on the detector. The receiver was a HgCdTe detector with a working cut-off wavelength of 6 μm. The optical area was 1 × 1 mm^2^, the acceptance angle was 36°, and the time constant was 50 ns. The optical axis distance between the laser and detector was around 45 mm, and the diameter of the corner mirror was 50.8 mm. When the corner mirror was 8 m away from the laser source, the incident light and reflected light were at the edge of the corner mirror. A sawtooth wave with a frequency of 5 Hz and a sine wave with a frequency of 31.4 kHz were both produced by the field programmable gate array (FPGA) module. The modulation signal was loaded into the laser controller to modulate the laser source. The signal received by the detector was input to the FPGA module, and the second harmonic signal was output after the phase-locking operation. The second harmonic signal was gathered by a data collection card and shown on the computer software.

To determine the gas concentration, the gas detection system based on TDLAS requires calibration using standard concentration gas. One easy way to set up standard gas concentrations is through static gas distribution. Calculations can be performed to determine the gas concentration when a specific amount of gas is supplied to a container with a known volume. A gas chamber with an inner diameter of 150 mm and a length of 1 m was put in the optical path. An inelastic gas bag was used to contain the CO gas after it was discharged from the high-pressure gas cylinder at the same pressure as the atmosphere. After the air chamber was evacuated, the CO gas was retrieved from the air bag and injected into it. After that, nitrogen was added to the gas chamber to bring the pressure there closer to that of the atmosphere.

## 4. Results and Discussion

The calibration concentration was determined by detecting CO in the gas chamber using an electrochemical gas detector. The electrochemical gas detector had a detection range of 1–1000 ppm and a detection accuracy of 1 ppm. As shown in Figure 3, CO was fed into the gas chamber in amounts of 0.2 mL, 0.5 mL, 1 mL, 2 mL, and 4 mL. The measured concentrations were 12 ppm, 27 ppm, 52 ppm, 112 ppm, and 217 ppm, respectively. The actual concentration of the gas was rather close to the theoretically calculated concentration, and the error may have resulted from a volume error in the injected gas. Through concentration calibration, the concentration of CO in the open optical path can be directly obtained.

Different CO concentrations in the gas chamber were measured using the CO detection system based on TDLAS. In Figure 4, the detection results were displayed. The amplitude of the second harmonic signal (ASH) grew linearly with the gas concentration, confirming the concept. Calculations revealed that the ratio of ASH to the gas concentration was 4.431 × 10^−3^/ppm. The lower detection limit of CO in the gas chamber was approximately 12 ppmm. (1 ppmm means that the concentration of 1 ppm gas was distributed in the length of 1 m.) CO gas at a concentration of 240 ppb could be detected if a multipass absorption cell with an optical path length of 50 m was utilized. The CO gas concentration corresponding to the amplitude of various second harmonics of the gas detection system might be determined using the ASH of CO gas with varied concentrations in the gas chamber as the standard value. The signal-to-noise ratio can be improved, low-frequency background noise can be suppressed, and the limit of detection concentration can be lowered compared with the laser direct absorption detection technique. It can detect CO with low concentration; thus, gas leakage can be found at an early stage to keep people safe.

The scanning period of the laser modulation signal was 200 ms. By continually collecting the detection signal for 50 times, it was possible to achieve the stability of the gas detection result within 10 s. Figure 5 displayed the stability of the ASH for gases at various concentrations. The relative standard deviations of CO detection signals with concentrations of 12 ppm, 27 ppm, 52 ppm, 112 ppm, and 217 ppm were 0.019, 0.029, 0.024, 0.02, and 0.017, respectively. The measurement results of the CO detection system based on TDLAS were stable because the low-frequency background noise could be suppressed. The detection result is stable and reliable; therefore, this system can work for a long time to detect CO gas in the environment.

The silicon slices used for the gas cell windows had a thickness of 5 mm, and their transmittance at 4.625 μm was approximately 96.3%. Through the gas cell, about 14% of the laser’s power was lost. The ASH was proportional to the gas concentration and the laser emission power. According to the calibration of the standard concentration gas in the gas chamber, the ratio of the ASH in the open optical path to the gas concentration was 5.152 × 10^−3^/ppm (4.431 × 10^−3^/ppm/(1–14%)), and the minimum detectable concentration was about 10.32 ppmm (12 ppmm×(1–14%)). CO gas was sprayed in the optical path directly, and the measured second harmonic signal was shown in Figure 6a. The ASH was 2.638, and the concentration of CO gas in the optical path was approximately 510.6 ppmm. The concentration of CO in the optical path within two minutes was shown in Figure 6b. The detection result increased significantly right away after the gas was sprayed, and the concentration fluctuation during the gas diffusion was not significant. The CO content in the optical path immediately decreased as soon as the ventilation fan was turned on. The concentration of CO was much lower after one minute of ventilation, although there was still some CO gas present in comparison to before degassing. The diffusion of CO gas in the optical path was generally uniform because of the air movement. This system can detect the concentration of CO in the environment and give an alarm when the concentration exceeds the standard value. The detectable concentration is low, so that the danger can be detected in time. In confined spaces, timely ventilation can quickly reduce CO in the environment, thereby reducing the impact on health.

## 5. Conclusions

The laser detection of CO usually uses a long optical path gas chamber to improve the detection sensitivity, and there are few studies on the telemetry system. A CO gas telemetry system based on TDLAS was constructed using a 4.625 μm mid-infrared ICL with a high CO absorption coefficient in order to increase the detection sensitivity. The standard concentration of CO gas was prepared by the static gas distribution method. The standard concentration gas was calibrated in the gas chamber. The results of the gas detection were stable and reliable. The detection distance for CO gas in the open optical was roughly 8 m. The lower limit of detection concentration was approximately 10.32 ppmm, according to the calibration of standard concentration gas in the gas chamber. The gas detection distance can be extended by increasing the diameter of the corner mirror and the laser intensity.

## Figures and Tables

**Figure 1 ijerph-19-12828-f001:**
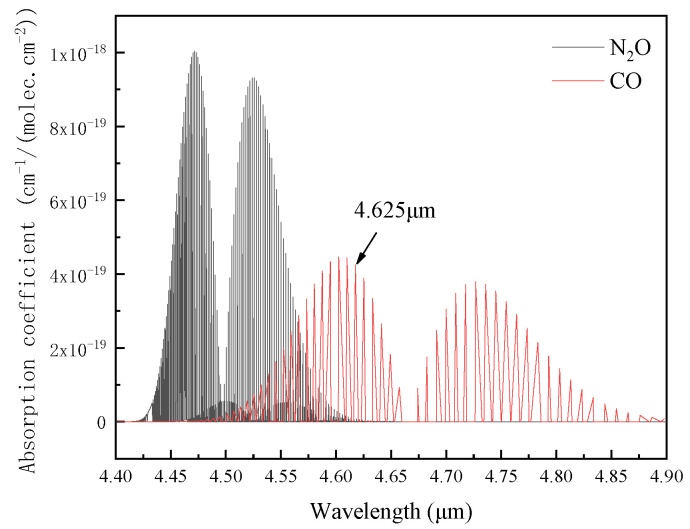
Absorption spectra of CO and N_2_O.

**Figure 2 ijerph-19-12828-f002:**
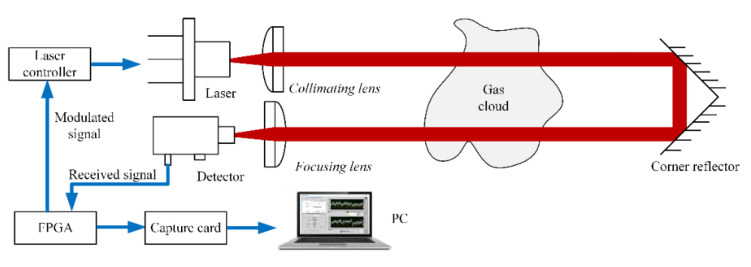
Gas remote detection system based on TDLAS.

**Figure 3 ijerph-19-12828-f003:**
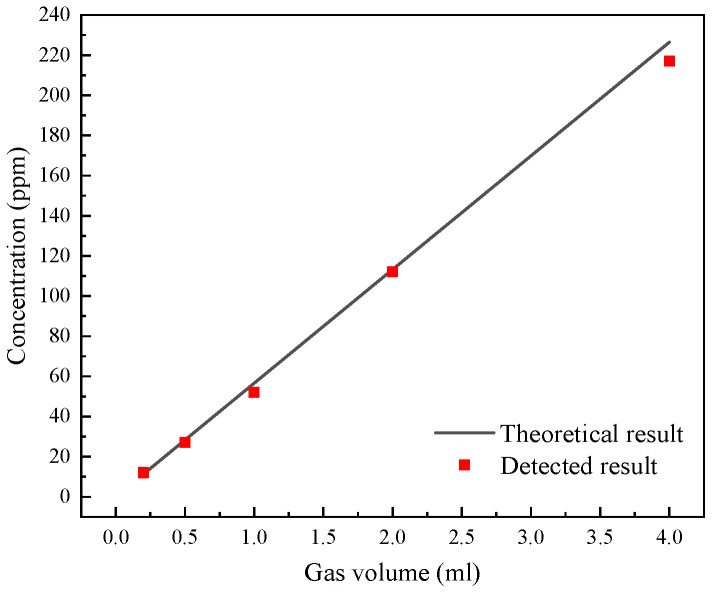
CO concentration calibrated with an electrochemical gas detector as a function of CO gas volume added into the gas chamber.

**Figure 4 ijerph-19-12828-f004:**
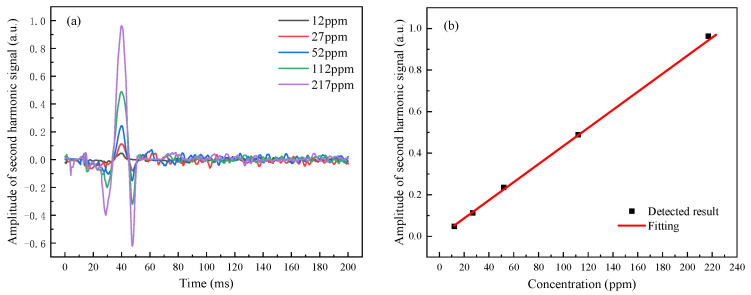
Detected result of CO with different concentrations in the gas chamber based on TDLAS: (**a**) waveforms of the second harmonic signal; and (**b**) amplitudes of the second harmonic signal.

**Figure 5 ijerph-19-12828-f005:**
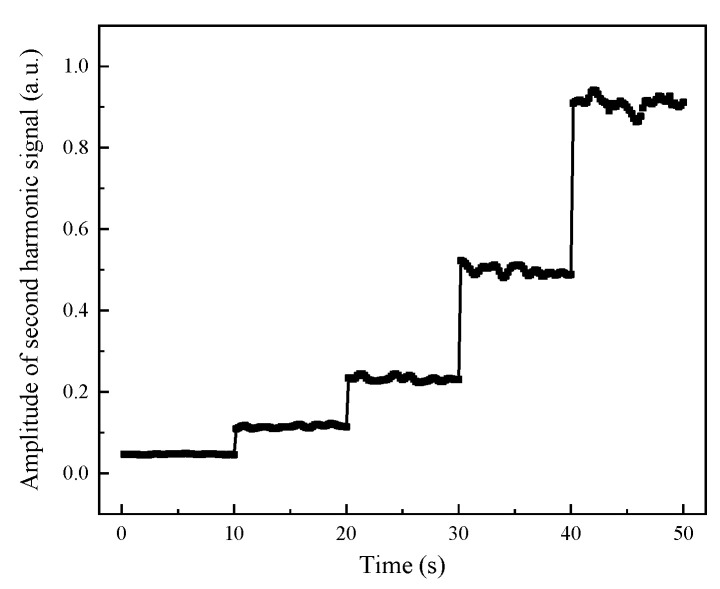
The fluctuation of ASH with different concentrations of CO in the gas chamber.

**Figure 6 ijerph-19-12828-f006:**
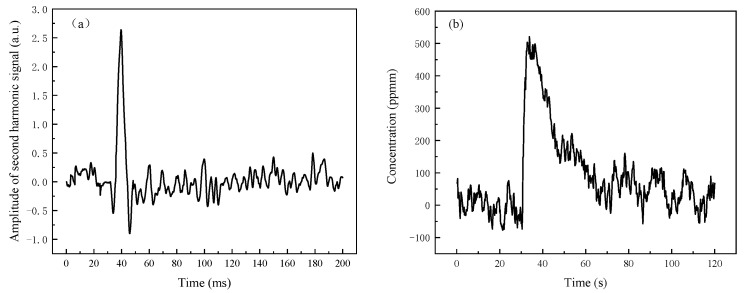
Remote detection result of CO based on TDLAS in the open optical path: (**a**) waveforms of the second harmonic signal; and (**b**) fluctuations of the concentration.

## Data Availability

Not applicable.

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
