# Peer review of "CO Detection System Based on TDLAS Using a 4.625 μm Interband Cascaded Laser"

_ijerph, 2022, doi:10.3390/ijerph191912828_

Round 1

Reviewer 1 Report

The authors of this research proposed an alternative to the traditional laster methodology for the detection of CO in an open environment. I think that this research is of interest and could suppose an important advance in the detection of this compound with dangerous repercussions to health. However, the article presents some important lacks that have to be corrected before being published.

Format:

-Please take care about brackets, sometimes you include a space before and sometimes not. Please homogenize it.

-Include the number of references after the authors. Line 50: Chen et al. [14]; line 58: Shao et al. [16].

-Line 160, 161: correct the comma after the numbers, for example, 0.01, not '.

-Review references, the journal name have to be in IUPAC abbreviation.

Introduction:

-Line 27: Delete will and include "produceS"

-Line 48: Introduce the disadvantages of the technique in a more clear way. For example "However, some of the disadvantages associated with this technique are a low signal-to-noise ratio, etc... Many of them can be improved with TDLAS use.

Sections 2 and 3 should be subsections of a "Materials and Methods section". In addition, the paragraph from lines 122 to 134 should be included in the materials and methods.

The discussion of the results is significantly poor. You must compare your results with the currently used methods, you have to prove the advantages of using your alternative over the other. It is not enough to present your results.

For last, the authors have included in the introduction that temperature plays a key role in CO production and detection. Why they have not included this variable to measure the CO concentration?

Reviewer 2 Report

This work demonstrated a tunable diode laser absorption spectroscopy (TDLAS) system working at the CO absorption window around 4.625 µm for CO gas detection. The minimum detectable concentration of 10.3 ppmm was achieved when the length of the optical path is 8 m. This work is important and novel. However, the writing needs to be improved to reach the standard of academic publications. In my opinion, this work is suitable to be published in the International Journal of Environmental Research and Public Health after a minor revision.

1.     All the abbreviations need to be defined in the main text even some of them were already defined in the abstract.

2.     I recommend adding a space between number and unit, e.g., “10.3ppmm” to “10.3 ppmm”, to be consistent with most scientific publications.

3.     I recommend changing the axis label style from “distance/mm” to “distance (mm)” to avoid confusion.

4.     The TDLAS is a very mature technology that has been developed since the 1970s. The references in the introduction part are all very recent. The authors need to cite some most influential works in this field, which can guide the readers to have more understanding of this field.

5.     The  in eq. 4 should be . Also, it is not immediately clear how eq. 4 can be derived from eq. 1-3. The authors need to provide details or references such as the one listed below to help readers understand the derivation. Further, all the symbols need to be explained immediately after the equation in which they were used.

P. Werle, “A review of recent advances in semiconductor laser based gas monitors,” Spectrochim. Acta Part A Mol. Biomol. Spectrosc., 54, 197–236, (1998).

6.     In the description in lines 101 and 102, “At constant temperature, the laser source was tuned around 15 nm by current, with a linewidth of less than 3MHz.”, the units are better to be all in nm or Hz.

7.     Nearly all the figure captions are sloppy and lack accuracy. For example, the caption “concentration of gas distribution” of fig. 3 may be changed into “CO concentration calibrated with an electrochemical gas detector as a function of CO gas volume added into the gas chamber”

8.     Since the “amplitude of second harmonic” were so frequently used, maybe it is better to give it an abbreviation so that many sentences would be more readable.

9.     The standard deviations (SD) reported in lines 160 and 161 are meaningless since the arbitrary unit does not give readers an intuition of the fluctuation. It is better to use relative standard deviation (SD/mean).

10.  In line 170, 10.3 ppmm was given as the detection limit. The authors need to explain how this is obtained.  

Round 2

Reviewer 1 Report

I consider that the changes made by the authors solve all the issues that I found previously in this article, so I accept it for its publication.